

# Bioturbation by wild boar increases the stability of forest soil carbon

Axel. Don[1], Christina Hagen[1], Erik Grüneberg[2], Cora Vos[1],

[1]Thünen Institute of Climate-Smart Agriculture, Braunschweig, 38116, Germany
[2]Thünen Institute of Forest Ecology, Eberswalde, 16225, Germany

*Correspondence to*: Axel Don (axel.don@thuenen.de)

**Abstract.** Most forest soils are characterised by a steep carbon gradient from the forest floor to the mineral soil, indicating that carbon is prevented from entry into the soil. Bioturbation can help incorporate litter-derived carbon into the mineral soil. Wild boar are effective at mixing and grubbing in the soil and wild boar populations are increasing in many parts of the world. In a six-year field study, we investigated the effect of wild boar bioturbation on the stocks and stability of soil organic
carbon in two forest areas. Regular bioturbation mimicking grubbing by wild boar was performed artificially in 23 plots and the organic layer and mineral soil down to 15 cm depth were then sampled. No significant changes in soil organic carbon stocks were detected in the bioturbation plots compared with non-disturbed reference plots. However, around 50% of forest floor carbon was transferred with bioturbation to mineral soil carbon and the stock of stabilised mineral-associated carbon increased by 28%. Thus, a large proportion of the labile carbon in the forest floor was transformed into more stable carbon.
Carbon saturation of mineral surfaces was not detected, but carbon loading per unit mineral surface increased by on average 66% in the forest floor due to bioturbation. This indicates that mineral forest soils have non-used capacity to stabilise and store carbon. Transfer of aboveground litter into the mineral soil is the only rate-limiting process. Wild boar can help to speed up this process with their grubbing activity.

**1 Introduction**

The stability of soil organic carbon (SOC) strongly depends on its association with soil minerals (Lehmann and Kleber, 2015; von Lützow et al., 2006). Organic carbon (C) that enters forest soil as aboveground litter is retained in an organic layer, with little opportunity for long-term stabilisation. Carbon in the organic layer is vulnerable to disturbances such as wildfire, windthrow or forest clear-cutting, since it is present in non-stabilised organic matter and exposed at the soil surface
(Zakharova et al., 2014; Jandl et al., 2007). The organic layer in German forests stores on average 19 Mg C ha-1, which may reduce the available amount of SOC that enters the mineral soil below an can be stabilised (Grüneberg et al., 2014). Translocation of organic layer C into the mineral soil is necessary for its stabilisation, but incorporation of dissolved organic matter (DOM) and particulate organic matter (POM) from the forest floor into the mineral soil is slow, as demonstrated by studies using isotope techniques (Fröberg et al., 2007a;Hagedorn et al., 2004;Arai et al., 2007). Besides downward
movement of dissolved organic matter, bioturbation is the main process that brings organic matter from the forest floor into the mineral soil. Bioturbation by earthworms has been found to be very effective in translocating litter C from the surface to mineral soil (Don et al., 2008). The invasion of European earthworms into some North American forests has led to complete disappearance of the forest floor due to bioturbation, but most of this forest floor C was found in the underlying mineral soil




(Alban and Berry, 1994). Most earthworms are restricted to non-sandy, non-acidic soils (Curry, 2004), while bioturbation by other animals such as termites, small rodents and ants can be found in many soils irrespective of soil texture (Wilkinson et al., 2009). However, wild boar (*Sus scrofa*), also called feral pigs or wild pigs, are most important for bioturbation in forests. This species is native to Eurasia, but has dispersed over all continents except Antarctica due to its adaptability, high

reproduction rate and secretive nature (Barrios-Garcia and Ballari, 2012). Wild boar is thus one of the most widely distributed species of mammal in the world. After almost becoming completely extinct in many parts of Europe, such as Great Britain, Scandinavia, Russia and large parts of Germany, by the beginning of the 20th century, wild boar populations have experienced a tremendous increase globally in recent decades, re-invading large parts of their former territory and beyond. The number of hunted wild boars has increased by over 10-fold in Germany, from 50 000 animals in 1950 to more

than 600 000 animals today (Arnold et al., 2015). The wild boar populations in France, Italy, Eastern Europe, North America and Asia are also increasing, most likely due to climate warming and changes in agricultural management providing more food, high reproduction rates and adaptation to a wide variety of habitats (Geisser and Reyer, 2005). Moreover, increased frequency of fructification of deciduous forest trees, such as beech and oak, has helped to increase food availability for wild boars and insufficient game management (Servanty et al., 2009). Wild boar can negatively affect agricultural land by

destroying grassland vegetation cover and feeding on crops such as maize. In contrast, wild boar may have mainly positive effects in forests, e.g. by encouraging natural regeneration of trees that require mineral soils for germination (Bruinderink and Hazebroek, 1996), enhancing tree growth by grubbing  (Lacki and Lancia, 1986), spreading plant seeds and fungi (zoochory), suppressing some pest (invertebrate) species and removing carrion. The vast majority of ecological studies on wild boar agree on the significant impact of the species on plant and animal communities and the ecosystem. However, there

is a surprising lack of quantitative data on the environmental impact of wild boar (Massei and Genov, 2004).

Wild boar are effective soil disturbers, since they can easily rifle with their snout to 5 to 15 cm depth in the soil, an activity called grubbing or rooting (Kotanen, 1995). Thus, the forest floor becomes mixed with the mineral soil (Singer et al., 1984). Wild boar are omnivorous and obtain a considerable proportion of their diet by grubbing in the soil, seeking for food such as acorns or other fruits from forest trees, bulbs, annelids, molluscs and mushrooms. However, soil disturbance through

bioturbation or tillage is reported to cause SOC losses via enhanced organic matter mineralisation (Franzluebbers, 1999; Kristensen et al., 2000). By breaking up aggregates and aerating the soil, microbial SOC turnover may be stimulated. Disturbance by tillage of native sward in North American prairie grassland causes rapid SOC loss (Mann, 1986). Soil disturbance by bioturbation through wild boar grubbing may have a significant impact on SOC and nutrient cycling in forest ecosystems, since in temperate forests most nutrients and SOC are stored in the forest floor and the upper mineral soil

(Wirthner, 2011). However, the impact of these large mammals on forest SOC and nutrient cycling and stocks has largely been neglected. Disturbance by tillage, harvesting, storm damage, fire, drought or insects is frequently mentioned in forest soil inventories and studied (Overby et al., 2003; Nave et al., 2010), but wild boar disturbance is generally not reported (Schulp et al., 2008). While numerous studies demonstrated the influence of forest management and disturbances on SOC



stocks, such as tree species selection (Vesterdal et al., 2013), thinning (Jurgensen et al., 2012), harvesting (Nave et al., 2010) and liming (Melvin et al., 2013), wild boar disturbance effects have rarely been investigated.

Mixing of soil horizons and incorporation of energy-rich organic material into the mineral soil can stimulate microbial activity and respiration, and therefore may enhance decomposition (Mallik and Hu, 1997; Risch et al., 2010). Thus, the labile

organic C of the forest floor may be lost with invasion and further expansion of wild boar populations, with negative consequences for forest SOC stocks and side-effects for the global C cycle and climate change. Detection of wild boar effects on SOC is difficult, due to the large inherent spatial variability of forest floor and mineral soil SOC stocks and the additional variability caused by wild boar grubbing (Wirthner, 2011). A dedicated experimental design and sufficiently large samples are required to quantify changes in SOC due to bioturbation. In addition, reference sites that have definitely never

been grubbed by wild boar in the past are required. The aim of this study was to quantify the effect of wild boar bioturbation on SOC stocks and the degree of stabilisation of SOC in coniferous and deciduous forests. The consequences of wild boar grubbing on organic matter stocks in forests were quantified.

## 2 Material and Methods

### 1.2 Study sites

Two study areas were selected and a total of 24 research plots scattered within these areas were established. The first area is located 3 km north-west of the city of Braunschweig (coordinates: 52°17'N, 10°26'E), in a region with mean annual temperature of 8.8°C and mean annual precipitation of 620 mm. The soil is this area is a Lamellic Luvisol developed on periglacial loess and sand deposits with a moder organic layer. The forest in the area is diverse, comprising deciduous forest dominated by beech and oak, coniferous forest dominated by spruce and Douglas fir and mixed forest. The second area is

located close to the city of Eberswalde, around 50 km north-east of Berlin (52°52'N, 13°49'E). Mean annual temperature in this area is 8.9°C and mean annual precipitation is only 520 mm. The soil is a Dystric Cambisol derived from Pleistocene sand deposits with a moder organic layer. The forests in this area are pure pine. Both study areas were fenced in order to exclude wild boar and these fences were successful throughout the study period in the Eberswalde area and in the early part of the period in the Braunschweig area, where wild boar have recently invaded the forests. However, regular inspection

revealed no impact or disturbance by wild boar in the study plots.

### 1.3 Treatments at the research plots

In the Braunschweig area, 18 study plots (six in deciduous forest, six in mixed forest and six in coniferous forest) were established and marked in early spring 2011. Due to a windthrow event, one coniferous plot was lost. At the same time, six research plots, all in coniferous forest, were established in the Eberswalde area. The plots at both sites consisted of two

subplots, each measuring 2 m by 4 m, located directly adjacent to each other and positioned at sufficient distance from trees or other objects, such as ditches or the forest edge that could influence the forest floor or soil. One subplot was used to



simulate bioturbation by wild boars, using poles with one end shaped to represent a wild boar tusk (hoes were also used in Eberswalde plots due to very dense grass sward). This bioturbation activity was conducted manually by 2-3 people, who plunged the tool through the forest floor and the upper mineral soil for 10-20 minutes per plot, to mimic the action of wild boar tusks. The resulting bioturbation pattern was random and the results very visually similar to forest soil patches disturbed

by wild boar. With this treatment we mimicked bioturbation by wild boar but not removal of material on which wild boar are feeding on. Also the admixture of dung from wild boar is not included. This allows distinguishing the bioturbation effect from other effects of wild boar grubbing. The bioturbation treatment was performed once per year, always in spring, for a period of six years, in order to examine longer-term effects of bioturbation on forest soils. The second subplot in each plot was left undisturbed and served as a reference plot.

**1.4 Soil sampling**

Soil sampling was conducted in late summer 2016 in both areas. In order to obtain representative soil material, very large samples were collected, using a 25 cm x 25 cm metal frame, with three replicates per subplot (only two replicates in the Eberswalde area, since all six plots were in close proximity). In the reference subplots, two organic layers of the forest floor (L-layer and O-layer) and three depth increments in the mineral soil (0-5 cm, 5-10 cm, 10-15 cm) were sampled. In order to

make the sampling in the disturbed plots comparable in those plots, we sampled mass equivalent as proposed by Ellert and Bettany (1995) (Fig. 1). For each plot, we used the mean weight of the three replicates of the organic layer (combined sample L- and O-layers) and the three mineral soil depth increments determined in the field in order to sample equal masses (Fig. 1). By sampling equal masses instead of equal depth increments, we accounted directly for the changes in bulk density due to wild boar grubbing. We assumed that the moisture content was similar in the reference plot and the disturbed plot, but

corrected deviations from this assumption afterwards in the dataset in order to calculate mass-corrected SOC stocks based on dry matter. The sampling of forest floor layers and mineral soil from the same spot (metal frame) ensures that deviations in separation of forest floor and mineral soil do not affect estimates of total SOC stocks (Don et al., 2012). In total, 567 samples were obtained, with a mean weight per sample of 3.8 kg for the mineral soil and 0.6 kg for the organic layer samples (Fig. 1).

**1.5 Sample preparation and analysis**

All samples were dried at 60°C and organic layer samples were cut (<1 mm) and homogenised in an electric mill. Mineral soil samples were sieved through a 2-mm mesh and a subsample was finely milled for further analysis. For each sample, dry fine soil mass was determined gravimetrically and the C and nitrogen (N) concentration was measured using dry combustion (TrueMac, Leco, USA). Using these parameters, it was possible to calculate SOC stocks according to Poeplau et al. (2017). Inorganic C was not present in any of the samples. For composite samples of all sites in the Braunschweig area and sites 1

and 4 of the Eberswalde area, pH was determined in 1 M KCl solution with a soil:solution ratio of 1:5. Soil texture classes in composite mineral soil samples (5-10 cm depth) were estimated for each plot using the texture-by-feel technique (Vos et al., 2016).



In order to characterise the degree of stabilisation of SOC, density fractionation was performed for composite mineral soil samples of 0-5 cm depth and 5-10 cm depth derived from a mixture of all field replicates. Using the protocol from Golchin et al. (1995), SOC was divided into three fractions: free particulate organic matter (fPOM), occluded particulate organic matter (oPOM) and mineral-associated organic matter (MOM). In addition, mineral-associated C on minerals mixed into the

organic layer by bioturbation was determined, using a similar protocol. In brief, 4 g (0-5 cm depth samples) or 13 g (5-10 cm depth samples) of soil were suspended in a 1.6 g cm$^{-3}$ polytungstate solution (soil:solution ratio 1:10). After ultra-sonication of the samples, minerals were centrifuged and floating particulate organic matter was removed, washed with deionised water and dried at 70°C.

### 1.5 Statistical analysis

The recovery rate of soil mass and C content in the fractionation procedure was calculated by adding up the C content of the individual fractions and comparing the total with the original C content at the start of the fractionation.

Statistical analysis was performed using the software R version 3.3.0 (R Core Team, 2016). Descriptive statistics (including means and standard deviations) were calculated for the C stocks in each area, in plots and in individual layers in each plot, and for the C stock distribution among the different soil fractions.

## 2 Results

### 2.1 Soil organic carbon stocks in forest floor and mineral soil after bioturbation

The SOC stocks in the forest floor and the mineral soil (0-15 cm depth) did not change significantly due six years of the wild boar bioturbation treatment (Fig. 2A). Total C stocks in the forest floor and mineral soil (0-15 cm depth) were 75±10 Mg ha$^{-1}$ (mean ± standard error) in the wild boar plots and 77±9 Mg ha$^{-1}$ in the undisturbed reference plots. However, the proportion

of mineral SOC stocks increased by 26 % and, correspondingly, forest floor C stocks decreased by 40 % due to bioturbation. Without bioturbation, 51 % of total SOC (down to 15 cm depth) was stored in the forest floor and 49 % in the mineral soil. With wild boar bioturbation, SOC was redistributed, leaving only 29 % of total C in the forest floor and 71 % in the mineral soil. In 22 out of 23 plots, the mineral SOC stock was increased by bioturbation (Fig. 3). However, the variability of SOC between the different plots was large also within the same region and forest type. The C gradient after bioturbation was less

steep, with an average SOC content of 16 % in the organic layer compared with 33-49 % in the organic layers of the non-disturbed reference sites (Fig. 2b). Thus, mixing of mineral soil into the forest floor decreased the C content in the forest floor but increased the C content in the upper mineral soil. However, the wild boar bioturbation treatment was shallow: Below 5 cm depth, effects on SOC were minor and no effect could be detected below 10 cm depth. Thus, our sampling was sufficiently deep to capture all bioturbation effects.



Differences in undisturbed SOC stocks between coniferous, mixed and deciduous forest were small and non-significant, with a difference of +3 %, -5 % and -0.2 %, respectively, compared with the average SOC stocks in all plots. In all three forest types, forest floor C stocks were reduced via bioturbation, by 37 % in coniferous forest, 46 % in mixed forest and 37 % in deciduous forest. In deciduous and coniferous forest, bioturbation effect on the forest floor was lower than in mixed forest.

This was attributable to greater thickness of the forest floor and a larger proportion of mineral soil SOC in deciduous and coniferous forest reference plots than in mixed forest plots. Overall, the reduction in forest floor C stocks due to wild boar bioturbation treatment was lower in the Eberswalde area (-25 %) than in the Braunschweig area (-45 %). However, changes in mineral soil C were greatest in Eberswalde, with SOC stocks doubling in the 0-5 cm depth layer. Plots in Eberswalde were covered with a grass sward that formed a dense root layer and this resulted in a particularly steep C gradient and prevents the

incorporation of aboveground litter into the mineral soil.

**2.2 Stability of soil carbon after bioturbation**

Bioturbation significantly changed the distribution between forest floor SOC and mineral soil SOC, as described above, and also changed the degree of stabilisation of SOC in the mineral soil. The proportion of particulate organic matter (fPOM and oPOM) increased significantly (p=0.04), from on average 52 to 60 % (Fig. 4a). This relative increase in POM proportion was

higher in the uppermost mineral soil layer (0-5 cm) than at 5-10 cm depth. Differences between forest types were small and non-significant, with a slightly higher POM fraction in coniferous reference forests (+59 %) than in mixed and deciduous forests (+45 % and +49 %, respectively) in the Braunschweig area.

However, even though the proportion of total stocks of MOM decreased with bioturbation, the MOM stocks in the mineral

soil did not change (Fig. 4b). This was due to the absolute increase in mineral soil SOC stocks with bioturbation. This increase in mineral soil SOC stocks (0-10 cm depth) can be attributed mainly to an increase in POM stocks with an increase by 58 % for fPOM and 43 % more oPOM. The MOM stocks did not change significantly in the mineral soil, but increased by 27 % in the total soil due to additional MOM in the forest floor. Minerals were mixed into the forest floor with bioturbation, resulting in a mineral-associated SOC stock of 5 Mg ha$^{-1}$ (Fig. 5). Thus, bioturbation did not cause detectable

C-losses but transferred floor C into the mineral soil where it is stored mainly as POM (Fig. 5). Within the six-year study period, we did not detect significant physical stabilisation of this transferred C in the mineral soil.

The quality of SOM, as indicated by the C/N ratio, changed with bioturbation, with decreased C/N ratio in the forest floor (Fig. 6). The ratio decreased from 38 to 28 in coniferous forest (average for total forest floor), from 31 to 23 in deciduous

forest and from 34 to 22 in mixed forest, indicating a higher fraction of transformed SOM in the disturbed forest floor. Mineral soil C/N ratio was only slightly affected by bioturbation (Fig. 6). This is in contrast to the large shift in fractions in the mineral soil, with a doubling of POM stocks.



### 2.3 Stabilising carbon on minerals with bioturbation

We found significant more adsorption of carbon on mineral surfaces due to bioturbation in the forest floor but no additional of SOC in the mineral soil. Minerals mixed into the forest floor in the Braunschweig plots were enhanced with C by 76% after bioturbation, from 24.2 g kg$^{-1}$ in the reference upper mineral soil to 42.7 g C kg$^{-1}$ in the forest floor (Fig. 7). The

increase was lower in the Eberswalde plots, with 29 % more C in the heavy fraction of the forest floor material.

## 3 Discussion

### 3.1 Disturbance by bioturbation does not reduce SOC stocks

Physical disturbance of soil, e.g. with tillage, is reported to enhance microbial activity and respiration and thus lead to SOC

losses (Sapkota, 2012; Franzluebbers, 1999; Kristensen et al., 2000). However, in a six-year study mimicking soil disturbance with wild boar grubbing in different forest types, we found no changes in total SOC stocks due to bioturbation, but detected redistribution of SOC from the forest floor to the mineral soil (Fig. 2). Similar results have been obtained for earthworm bioturbation, e.g. invasion of earthworms into North American forests has been found to cause rapid incorporation of the forest floor into the mineral soil, without considerable SOC losses (Bohlen et al., 2004). In a study re-

sampling six soil pits in northern USA hardwood forest (Tennessee/South Carolina) after invasion of wild boar, Singer et al. (1985) found that the forest layer mass was reduced by 60 % but with no change in organic matter content in the A-horizon. However, conclusions on changes in total SOC stocks could not be drawn in that study. Similarly, a Swiss study found that forest floor C was reduced by 40 % due to wild boar grubbing, but that total SOC stocks down to 30 cm depth were not significantly affected (Wirthner, 2011). However, in that study the reference sites may have been grubbed by wild boars

some years before sampling, although old grubbing patterns were not detected. A 14% loss of forest floor C has been found in a forest in the Netherlands, but total SOC stocks including the mineral soil are not reported (Schulp et al., 2008). The sampling design and methods in previous studies may not have been suitable for detecting SOC stock changes in disturbed soils. Studies that only report SOC content and no stocks, or only sample the forest floor, cannot be used to follow the fate of SOC after wild boar grubbing (Bruinderink and Hazebroek, 1996; Mohr et al., 2005;Moody and Jones, 2000). Lower SOC

content in the forest floor does not necessarily mean that SOC is lost, since mixing mineral soil into the forest floor can decrease SOC content without decreasing forest floor SOC stocks. Missing bulk density data and insufficiently shallow and small-scale sampling may also prevent appropriate conclusions being drawn from existing wild boar studies. The most difficult task, however, is separation of the forest floor from the mineral soil during sampling. Only combined sampling of both forest floor and mineral soil, as conducted in our study, can ensure that SOC stocks are determined correctly (Don et al.,

2012).





Our results also seem to contradict findings by Risch et al. (2010), who estimated an additional $CO_2$ source from Swiss forests due to wild boar grubbing of 50-98 Gg $CO_2$ per year. However, those values are based on soil respiration flux measurements that cannot be directly converted into SOC losses, since autotrophic $CO_2$ respiration could not be separated from heterotrophic respiration. Higher $CO_2$ respiration on grubbed plots can be explained by the reported higher fine root

5  density in grubbed plots, which was probably a consequence of the reported more dense ground vegetation. Thus, soil respiration measurements alone cannot be used to estimate SOC changes and to calculate the potential climate impact of wild boar grubbing. In an experiment on Hawaii, SOC stocks in the 0-10 cm layer were 12 % higher in plots with wild boars than in plots from which wild boar had been excluded for 7 to 19 years, supporting our finding of positive grubbing effects on total SOC stocks (Long et al., 2017).

Our results also question the claim that tillage or disturbance causes SOC losses. Tillage in comparison with no tillage always results in incorporation of crop residues into the mineral soil. Thus, all field studies comparing conventional tillage and no tillage report changes in the depth distribution of SOC, but many studies also report no changes in total SOC stocks (Baker et al., 2007; Hermle et al., 2008;Govaerts et al., 2009). Our study showed that organic matter can be mixed and

disturbed without C losses. As a consequence, the general effect of soil disturbance on SOC turnover and stocks may have to be re-evaluated. Any possible SOC loss due to enhanced mineralisation after soil disturbance may also be easily compensated for by the additional SOC that is stabilised. Bioturbation by wild boar brings together organic matter (from forest floor) and mineral particles such as phyllosilicates or oxides, which are essential in stabilising organic matter (Marschner et al., 2008;Eusterhues et al., 2003;Kaiser et al., 2002). In the present study, six years of frequent bioturbation

increased MOM by 28 %, but mainly in the forest floor (Fig. 5). Conversion of POM to stabilised MOM in the mineral soil may require longer time scales, and was thus not detectable.

The treatment mimicking bioturbation by wild boar mainly affected the upper mineral soil to 10 cm depth, but not below. This is the critical interface, with a large C-gradient between forest floor and mineral soil ranging from on average 33 % in

the O-layer to 6 % in the upper mineral soil (Fig. 1b). Thus, C-input with forest aboveground litter becomes trapped in the organic layer and, without bioturbation, there may be no C flux, apart from dissolved organic carbon, into the mineral soil that could contribute to building up SOC stocks (Fröberg et al., 2007b; Arai et al., 2007). The shallow bioturbation caused by wild boar can break up this C-flux barrier between forest floor and mineral soil and facilitate incorporation of aboveground-derived litter into the mineral soil. Consequently, we found that the C-gradient was clearly less steep in the bioturbation plots

than in the undisturbed reference plots (Fig. 1b).

## 3.2 Contact between organic layer carbon and minerals facilitates soil carbon stabilisation

Mixing forest floor carbon into the mineral soil by wild boar grubbing may enhance SOC stabilisation by adsorption onto mineral surfaces. However, we did not detect an increase in the mineral-associated fraction of SOC in the mineral soil within





the six-year study period (Fig. 5). Nevertheless, the minerals mixed into the forest floor adsorbed SOC, building up a new pool of stabilised SOC in the forest floor. Beside wild boar grubbing, more biological activity seems to be required to transform POM into stabilised MOM. The sandy soils with low pH investigated in this study (Table 1) may not provide sufficiently good conditions to facilitate this biological transformation in the given time frame.

Mineral soil is reported to store a limited amount of SOC stabilised as mineral-associated carbon (Six et al., 2002). The capacity of a soil to stabilise SOC largely depends on its mineralogy and texture, with clayey soils being able to store much more SOC in stabilised form due to their larger specific mineral surface area (Hassink et al., 1997). The soils in the present study are characterised by low specific mineral surface area due to their low clay content. In the reference plots, 48% of total

10 SOC was stabilised on mineral surfaces. However, the mineral-associated SOC content almost doubled within a short distance in the mineral soil (from 0-5 to 5-10 cm depth, see Fig. 4b), which indicates that, at least below 5 cm depth, the mineral surfaces are not carbon saturated. Thus, for SOC stabilisation the important issue is whether carbon reaches these mineral surfaces, rather than whether mineral surfaces are saturated or not.

The mixing of minerals and forest floor indicates that there is unexploited potential to adsorb and stabilise SOC on mineral surfaces. We found that forest floor mixed into the mineral soil almost doubled the C load (Fig. 7). Thus after bioturbation, 66 % more SOC per unit mineral surface was stabilised in the forest floor than in the upper mineral soil. This is surprising, since the soil at both study sites is sandy, with low specific surface area. However, in the extreme situation of minerals being surrounded by organic matter in the forest floor, more SOC becomes attached and thus stabilised onto mineral surfaces.

Thus, the mineral surfaces seem to be not limiting for SOC stabilisation but sufficient SOC is required in close vicinity to the mineral surfaces. In general, around one-third of the forest floor C was stabilised in the wild boar bioturbation treatment (Fig. 5). This SOC most likely originated from the forest floor and not from the mineral soil, as indicated by higher C/N ratio (18.3) compared with MOM in the mineral soil (C/N ratio 17.8). Moreover, the stock and the CN ratio of MOM in the mineral soil remained unaffected by the bioturbation treatment (C/N ratio 17.8 in reference plots and bioturbation plots).

**3.3 Areal extent of grubbing and vegetation feedbacks**

The area affected by wild boar grubbing can be extensive, with between 13 and 80 % of the soil surface affected in different forests inhabited by wild boar (Genov, 1981; Howe et al., 1981; Risch et al., 2010). From 27 to 54 % of Swiss deciduous forest area have been found to be disturbed by wild boar (Risch et al., 2010). In a study on a 70 km long forest-agriculture transect in southern Sweden, 1 to 6 % soil was found to be grubbed by wild boar each year, but with large inter-annual

variability (Welander, 2000). Similar results have been reported for Californian meadows, with wild boar grubbing an average of 7% of the study area annually (Krotanen 1995). Thus, the bioturbation frequency applied in the present study does not represent the average bioturbation frequency, but wild boar grubbing hotspots. Severely grubbed areas may extend for a hectare or more, but are typically composed of many small patches of a few square metres (Vallentine 1990, Krotanen

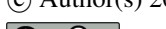

1995). Some patches may be preferentially grubbed compared with others, but the evidence on this is inconclusive. However, wild boar seem to prefer deciduous forest and damp soil over coniferous forest and grassland soil and dry soil (Welander, 2000). This is probably due to greater availability of feedstuffs such as acorns and beechnuts in deciduous forests and soil fauna in damp soil. With the global increase in populations of wild boar, forest soils are being increasingly grubbed, so wild boar-disturbed forest soils are now the rule rather than the exception.

The rather detrimental direct effects of wild boar grubbing on forest ground vegetation (e.g. mechanical damage, uprooting) can result in reduced plant cover. However, it is mainly the ground vegetation that is affected, and not the tree layer. In a Swiss study, the height of saplings (< 1 m) and plant species diversity did not differ between grubbed and non-grubbed plots (Wirthner, 2011). In the present study, we found temporarily enhanced ground vegetation at some plots in the Braunschweig area during the first years, with presence of the nitrophilous species Impatience parviflora indicating enhanced nitrogen availability (data not shown). Since all forests investigated were dominated by the tree layer, no significant effects of our wild boar bioturbation treatment on litter input to the soil were expected. Thus, the observed effects of the wild boar bioturbation treatment on SOC are due to the physical disruption, rather than other processes.

## 4 Conclusions

Wild boar is an invasive species in many parts of the world and their grubbing activities can be extensive. This soil disturbance caused a significant redistribution of SOC in the investigated forest plots with decreasing the organic layer. We provided strong experimental evidence of positive effects of a wild boar bioturbation treatment on SOC stability, whereby SOC in the forest floor is retained and transferred into the mineral soil without C-losses. Bioturbation does not cause SOC losses due to enhanced mineralisation, but in contrast helped to transform labile SOC into more stabilised SOC.

**Author contributions**

AD designed the experiment and AD and EG carried them out. All authors contributed to soil sampling and data interpretation. CH performed most soil analysis and CV performed the statistics. AD prepared the manuscript with contributions from all co-authors.

**Acknowledgements**

We thank Zohra Afshar, Viridiana Alcántara, Anita Bauer, Frank Hegewald, Angelica Jaconi, Christopher Poeplau, Annelie Säurich, Florian Schneider and Patrick Wordell-Dietrich for support during field sampling and during sample preparation in the laboratory, and Ines Backwinkel from the Central Laboratory of the Thünen Institute of Climate-Smart Agriculture for element analyses.

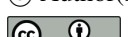



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



**Table1: Forest type, soil texture and soil pH in the plots at the Braunschweig (n=17) and Eberswalde (n=6) experimental sites.**

| Area | Plot no. | Forest type | Soil texture | | | pH |
|------|----------|-------------|-------------|-----------|-----------|------|
|      |          |             | Sand (%) | Silt (%) | Clay (%) |      |
| **Braunschweig** | 1 | Deciduous | 63 | 33 | 4 | 3.3 |
|      | 2 | Coniferous | 51 | 45 | 4 | 2.9 |
|      | 3 | Coniferous | 31 | 65 | 4 | 3.5 |
|      | 4 | Coniferous | 51 | 45 | 4 | 3.2 |
|      | 5 | Mixed | 31 | 65 | 4 | 3.1 |
|      | 6 | Deciduous | 51 | 45 | 4 | 3.1 |
|      | 7 | Deciduous | | NA | | NA |
|      | 9 | Coniferous | 51 | 45 | 4 | 2.9 |
|      | 10 | Mixed | 51 | 45 | 4 | 3.6 |
|      | 11 | Deciduous | 51 | 45 | 4 | 3.1 |
|      | 12 | Coniferous | 31 | 65 | 4 | 3.1 |
|      | 13 | Mixed | 65 | 33 | 4 | 3.2 |
|      | 14 | Deciduous | 51 | 45 | 4 | 3.0 |
|      | 15 | Mixed | 31 | 65 | 4 | 3.1 |
|      | 16 | Mixed | 51 | 45 | 4 | 3.0 |
|      | 17 | Deciduous | 51 | 45 | 4 | 3.6 |
|      | 18 | Mixed | 31 | 65 | 4 | 3.0 |
| **Eberswalde** | 1 | Coniferous | 80 | 17 | 3 | 3.1 |
|      | 2 | Coniferous | | NA | | NA |
|      | 3 | Coniferous | | NA | | NA |
|      | 4 | Coniferous | 92 | 5 | 3 | 3.1 |
|      | 5 | Coniferous | | NA | | NA |
|      | 6 | Coniferous | | NA | | NA |



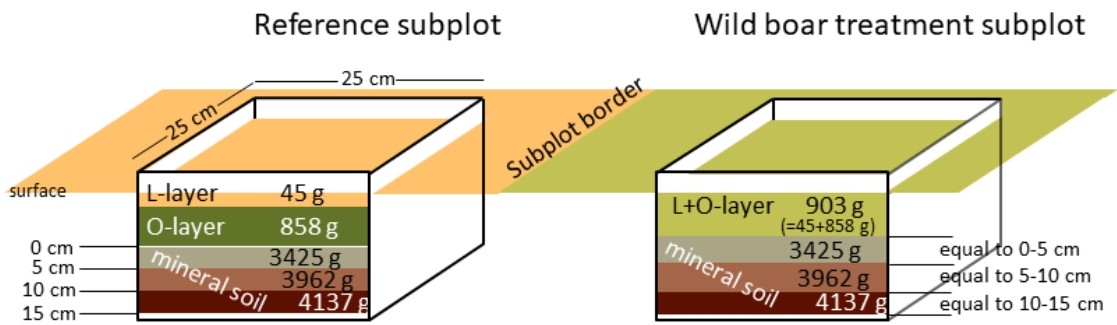

**Figure 1: Equal mass sampling of forest floor (L- and O-layer) and mineral soil (0-15 cm depth). Weights refer to the mean weight of the different layers and soil depth increments as obtained in the field.**

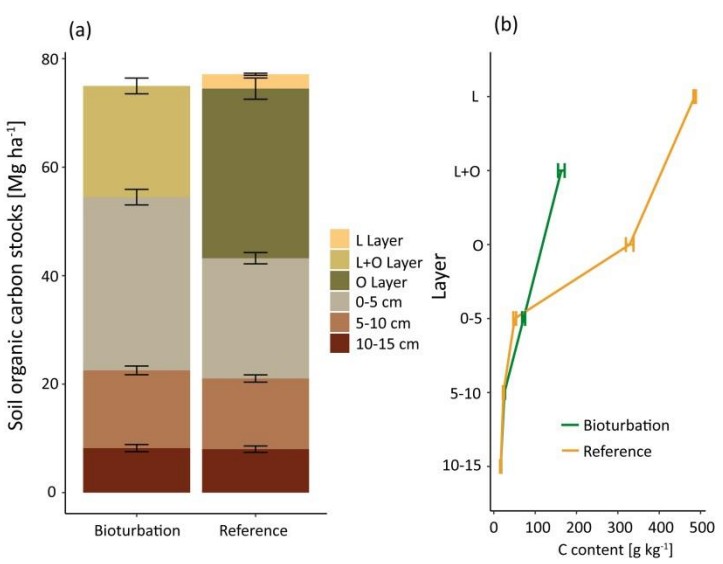

**Figure 2: (a) Mean soil organic carbon (SOC) stock [Mg ha$^{-1}$] distribution and (b) SOC content [g kg$^{-1}$] in different layers and soil**
10 **depth increments for plots in the treatment mimicking wild boar grubbing (Bioturbation) and in the undisturbed control sub-plots (Reference).**



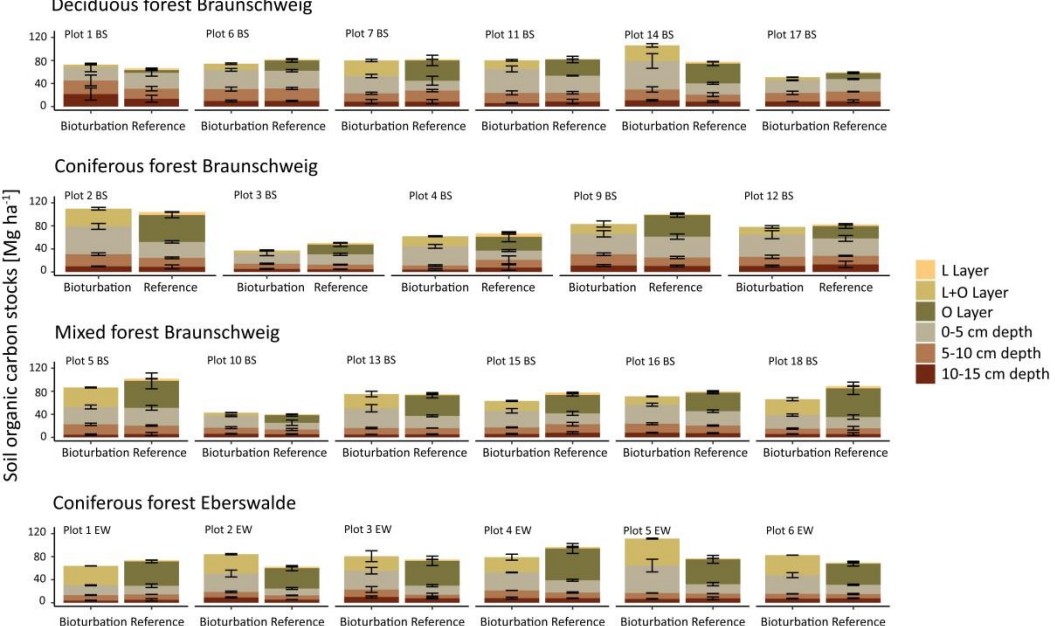

**Figure 3: Soil organic carbon (SOC) stocks [Mg ha⁻¹] in different layers in all 23 plots in the experimental areas Braunschweig (BS) and Eberswalde (EW).**

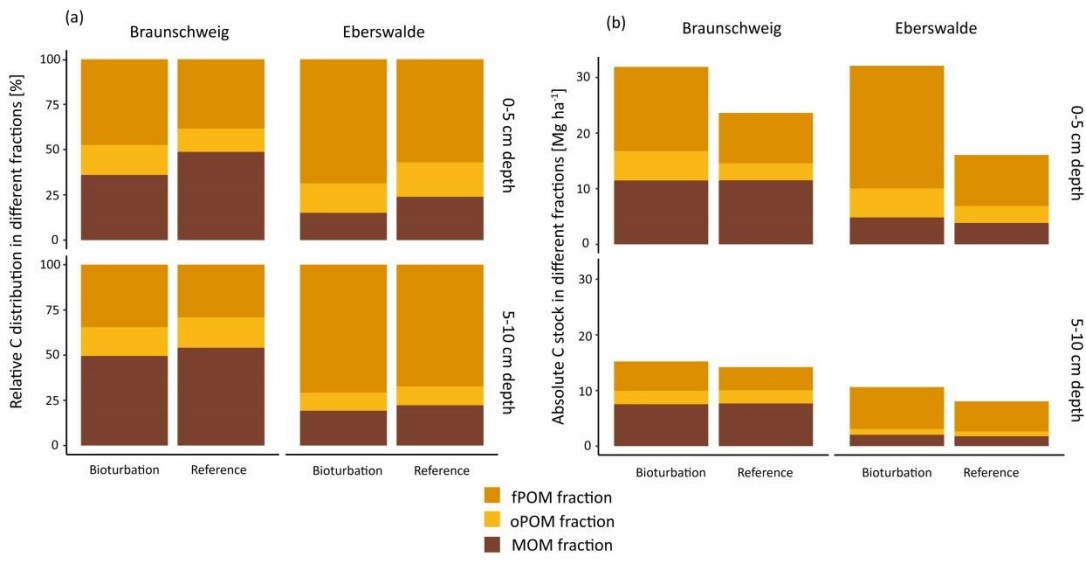

**Figure 4: (a) Relative and (b) absolute (b) distribution of soil organic carbon (SOC) fractions in the upper two mineral soil depth layers at the Braunschweig and Eberswalde experimental area after bioturbation and at reference sites.**



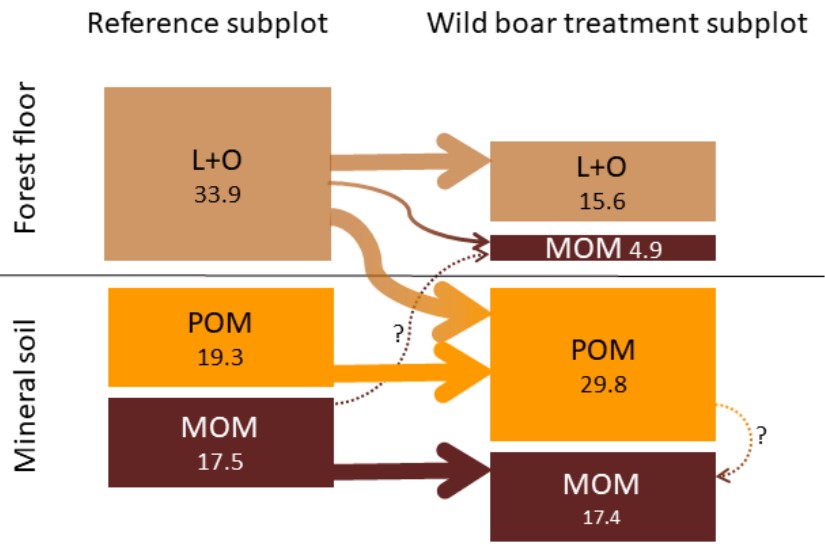

**Figure 5: Changes in different soil organic carbon (SOC) pools in the forest floor (L- and O- horizons) and the underlying mineral soil (0-10 cm depth) due to bioturbation. Average carbon stocks for all 23 plots [Mg SOC ha$^{-1}$]. POM = particulate organic matter, MOM = mineral-associated organic matter.**

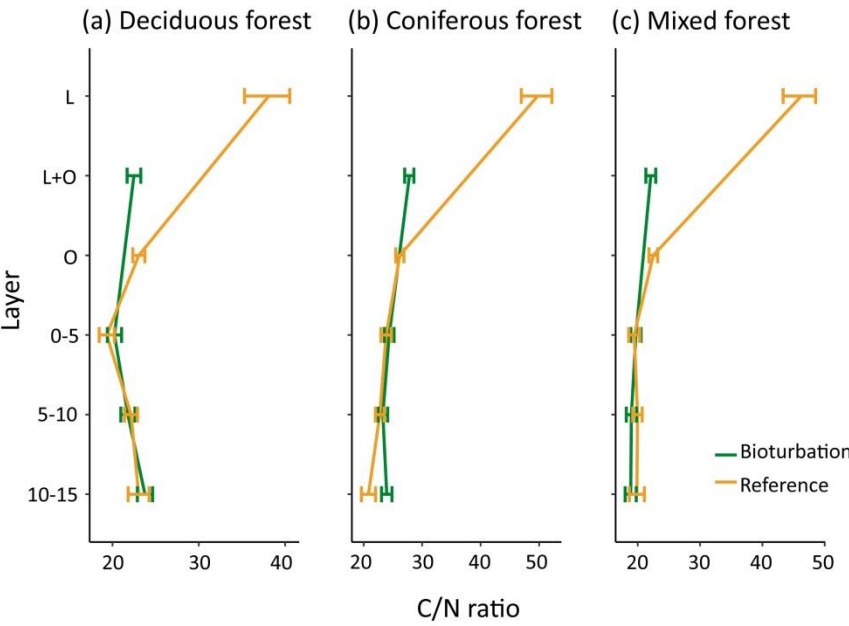

**Figure 6: Mean carbon:nitrogen (C/N) ratio of organic matter in different forest floor layers and mineral soil layers in disturbed and reference plots in deciduous (N=6), coniferous (N=11) and mixed forest (N=6). Bars indicate standard error.**





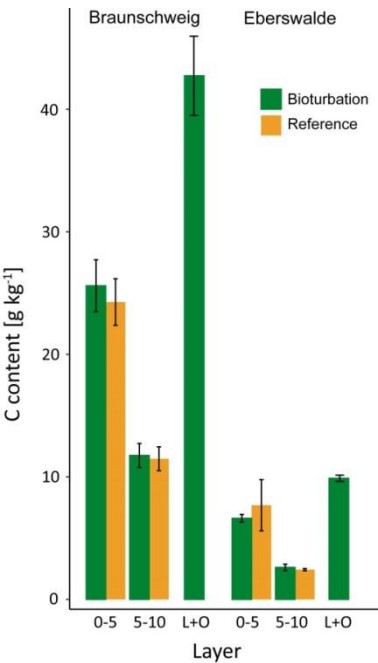

**Figure 7: Mean carbon (C) content of the heavy fraction in different soil depth increments at the Braunschweig and Eberswalde experimental areas. Bars denote standard error.**