# Peer review of "Simulated wild boar bioturbation increases the stability of forest soil"

_Biogeosciences, 2019_

## Referee Comment (RC1) · Anonymous Referee #1 · 21 May 2019

Review Report

Manuscript Number: bg-2019-113

Full Title: Bioturbation by wild boar increases the stability of forest soil carbon

Summary:

This study investigates the effect of simulated bioturbation by wild boar on forest soil carbon stocks and on soil C stability. Bioturbation was simulated by artificial soil disturbance down to the mineral soil. Total soil carbon stocks did not change after six years of regular soil disturbance. However, a major part of the litter layer carbon was incorporated into the mineral soil due to bioturbation. Accordingly, litter layer carbon stocks decreased and mineral soil carbon stocks increased following bioturbation. Moreover,

mineral-associated carbon increased due to soil disturbance. The authors suggest that mineral soils were not carbon saturated and have an unused capacity to stabilize and store more carbon. In conclusion, the authors claim that wild boar bioturbation may enhance (speed up) carbon stabilization in the mineral soil.

General comments:

Overall, I think this is a very nice study. Wild boar populations are increasing across Europe and their effects on soil carbon dynamics are still not fully understood. The manuscript is well written, the study design and measurements are sound and the results are interpreted in a good way. However, I have a number of concerns regarding the sampling procedure, the statistical analysis, and some of the figures. Please, find my specific comments in the following. I think after a revision the manuscript will make a valuable contribution to the research field and should be considered for publication in Biogeosciences.

Specific comments:

Title:

In my opinion, it should be added to the title that wild boar bioturbation was actually simulated.

Abstract:

P 1, L 7: Please rephrase 'can help'.

P 1, L 9: Add that wild boar bioturbation was simulated.

P 1, L 17: Please rephrase 'can help'.

Introduction:

P 1, L 30: I suggest either to replace 'the main process' by 'a major process' or to add an appropriate reference to that statement.

P 3, L 2: Please add the references of the studies which have investigated wild boar effects on soil carbon stocks.

P 3, L 11: Add that the effects of 'simulated' wild boar grubbing were investigated.

Materials and Methods

P 3, L 16, 17: I think there should be an 'a' before mean annual temperature and moisture.

P 4, L 14-19: It took me a while to understand the idea behind mass equivalent sampling, and why you applied it in this study. However, I'm still not sure if I properly understand it and I'm therefore a bit concerned if this procedure might affect the results. By sampling the same amount of soil per horizon and pit, I have the feeling you could underestimate potential C losses from bioturbation. For example, in the theoretical case, 50% of the LF horizon organic matter stocks would have been mineralized due to bioturbation, this sampling procedure would artificially 'refill' the missing amount of organic matter with organic matter from the next horizon (i.e. O horizon). Now, the O horizon is (artificially) smaller, but will be 'refilled' with soil from the next layer, and so on. At some point, material from a deeper layer which has not been sampled at the reference plot would be sampled to 'refill' the missing amount of organic matter. Thus, the actual amount of lost C would be underestimated. I might be completely wrong, but then I suggest to elaborate more in detail on the sampling procedure.

P 5, L 1-2: Was there only one composite sample per site, treatment and soil horizon? Please clarify.

P 5, L 14: Which statistical test did you use? What was you level of significance? How, did you account for nesting within sites? Please clarify. Results

P 5, L 17: Again, please indicate that bioturbation was simulated (here and elsewhere in the text).

P 5, L 18: Fig. 2a not 2A.

P 5, L 23: In my opinion it is not necessary to show the results of the individual plots. Thus I would suggest to move Fig. 3 to the supplements.

P 6, L 4: Instead of showing the individual plots (Fig. 3) I suggest to add a figure showing the bioturbation effect separate for the forest types.

P 6, L 8-10: This should be part of the discussion.

P 6, L 14: It is stated earlier that fractions were determined on composite samples only. How did you do statistics on that? Please clarify.

P6, L 15: Clarify that 'treatment effects' were similar among forest types. In the present form I first thought that e.g. POM fractions were similar among forest types.

P 6, L 19: is 'total stocks of MOM' correct? Or should it be MOM fraction? This reads a bit confusing. I would also suggest to refer to Fig. 4a here.

P 6, L 23-24: Please add the forest floor POM/SOM proportion and stocks to Figure 4. Although this results are included in Fig. 5, I think it would be more clear if you add it to Fig. 4.

P 7, L 4: Was there no mineral surface C in the forest floor of the reference plots? Did you measure it? Please clarify.

Discussion:

P 8, L 30: I guess it should be Fig. 2b not 1b.

P 9, L 10: Please cross-check figure reference.

Conclusion

P 10, L 16: Please, add that wild boar activity was simulated.

To put your results into a bigger context, I suggest to add some information/thoughts about potential long-term consequences.

Figures:

Figure 2: In the case horizons showed significant differences between treatments please indicate that by adding significance stars to the figure.

Figure 3: This figure should be moved to the supplements. Instead replace it with a barchart for each forest type. Add significance stars to the figure.

Figure 7: In the case horizons showed significant differences between treatments, please indicate that by adding significance stars to the figure. What happened to L+O of the reference plots?

---

## Referee Comment (RC2) · Anonymous Referee #2 · 3 Jun 2019

MSbg-2019-113: Bioturbation by wild boar increases the stability of forest soil carbon by Axel Don et al.

General comments:

Axel Don et al. have submitted an original, well written and very interesting draft to BG.

They investigated the effect of wild boar bioturbation on soil organic carbon (SOC) dynamics in a 6 year study in two forests of Germany (focusing on coniferous plots in one forest and coniferous + deciduous plots in the second forest), both on acidic and sandy soils.

The experimental design is nice and sound, the authors have manually simulated wild boar bioturbation (each year, which is a high frequency). They discuss in a clear way

the advantages and limitations of the chosen design. Yet, I think that in many sentences of the manuscript and also in the draft title, the fact that wild boar were not part of the game should be more clearly stated (see below my specific comment 1 on this topic).

SOC dynamics is studied in paired-plots (control vs. bioturbation) by focusing on SOC stocks (using equivalent soil mass for litter + 0-5/5-10/10/15 cm soil layers) and SOC physical fractionation to separate particulate SOC from mineral associated SOC (0-5/5-10 cm soil layers + C content of the mineral fraction of the O layer of bioturbed plots). The results show that SOC stocks were not affected by bioturbation but that the fate of litter SOC was affected by bioturbation: 1/ a part remains in the litter (as litter), 2/ a part is incorporated as particulate/light organic matter in the mineral topsoil layer, 3/ a part remains in the litter layer, but is associated to minerals.

The authors finally state that the part of the litter SOC that has been associated to minerals (in the litter layer) has been "stabilised" by bioturbation. I suggest that this statement on carbon "stabilisation" should be avoided. We indeed lack evidences regarding the residence time of mineral-associated SOC above the topsoil (i.e. in the litter layer; see below my specific comment 2 on this topic).

Specific comments:

1/ Wild boars were not involved in this study :)

I suggest to state more clearly in the title and in the text that bioturbation by wild boar was simulated. - "Simulated wild boar bioturbation..." for the title. - in the text this could be done for instance p5 line 17 "[simulated] wild boar bioturbation" and in many other sentences of the draft.

2/ Mineral-associated C in the litter layer (above-ground) cannot be called "stabilised" C

First, I would like to remind that in (mineral) soils, a large part of mineral-associated SOC is not stabilised. This has been clearly shown in e.g. long term bare fallows trials

where the fine soil fraction looses SOC at a relatively high rate. So transfering litter SOC to the mineral-associated SOC fraction does not mechanistically imply that all of it has been stabilised. A part of it may be stabilised if this transfer would have taken place in the mineral soil layer (i.e. below the soil surface). Indeed the mean residence time of mineral-associated SOC is generally higher that the one of the particulate organic matter SOC in mineral soil layers. But here the bioturbation transfer of SOC from litter to the mineral-associated C fraction occured in the litter layer (i.e. above the soil surface), where there is no evidence that this above-ground mineral-associated C would have a slower turnover than litter C from the F/H O layers.

This should be acknowledged in the manuscript.

- The title of the manuscript should be changed, avoiding the confusing term (and not properly measured for litter layers) "stability". The expression "increases mineral C loading" should be preferred and would better represent the findings of the study.

- The title of section 2.2 should be changed. Stability of SOC was not assessed, but "SOC distribution in physical soil fractions".

- The title of section 2.3 should be changed to "Associating C on minerals with bioturbation".

- The title of section 3.2 should be changed to "Contact [...] facilitates the association of litter C with minerals in the litter layer"

- The abstract/conclusion should be re-written: bioturbation has a positive effect on "C association to minerals" or on "mineral C loading in the litter layer", not on C stability, we do not know of this C is "more stabilised", it is more linked to minerals, and research on the turnover of mineral-associated C in the litter layer is therefore needed.

3/ The effect of wild boar on plant biodiversity and forest ecosystem C cycle is questionable

In the introduction section, the authors insist on the "mainly positive effects in forests"

(p2 line 15) of wild boar bioturbation. However, other studies are questioning this statement, presenting the effect of ungulate populations as :

- "jeopardiz[ing] forest regeneration process"

- "detrimental to the peculiarity of forest plant communities"

- leading to "lanscape-level biotic homogeneization" (see e.g. Boulanger et al., 2017 in Global Change Biology)

If forest regeneration process is actually jeopardized by wild boar invasions, then the fate of the global ecosystem C stock and cycle is not clear... This should be acknowledged in the manuscript.

4/ No positive grubbing effect on total SOC stocks were found

Please correct this mistake at p8 lines 9-10

Technical corrections:

p1 l26: "an[d]" ?

p5 l17: "due [to] six"

p7 l2: "significant[ly]

p8 l20: please replace "mainly" with "only"

p9 l16: please reverse "forest floor" and "mineral soil" : mineral soil mixed into the forest floor almost doubled the C load (not the opposite)

---

## Author Comment (AC1) · 17 Jun 2019

**Reply to Reviewer 1**
Summary:
This study investigates the effect of simulated bioturbation by wild boar on forest soil
carbon stocks and on soil C stability. Bioturbation was simulated by artificial soil disturbance
down to the mineral soil. Total soil carbon stocks did not change after six years
of regular soil disturbance. However, a major part of the litter layer carbon was incorporated
into the mineral soil due to bioturbation. Accordingly, litter layer carbon stocks
decreased and mineral soil carbon stocks increased following bioturbation. Moreover,
mineral-associated carbon increased due to soil disturbance. The authors suggest that
mineral soils were not carbon saturated and have an unused capacity to stabilize and
store more carbon. In conclusion, the authors claim that wild boar bioturbation may
enhance (speed up) carbon stabilization in the mineral soil.

General comments:
Overall, I think this is a very nice study. Wild boar populations are increasing across
Europe and their effects on soil carbon dynamics are still not fully understood. The
manuscript is well written, the study design and measurements are sound and the
results are interpreted in a good way. However, I have a number of concerns regarding
the sampling procedure, the statistical analysis, and some of the figures. Please, find
my specific comments in the following. I think after a revision the manuscript will make
a valuable contribution to the research field and should be considered for publication in
Biogeosciences.
**Reply:** We thank the reviewer for the very helpful comments and the appreciation of our study.

Specific comments:
Title: In my opinion, it should be added to the title that wild boar bioturbation was actually
simulated.
**Reply:** We agree and will change the title accordingly.

Abstract:
P 1, L 7: Please rephrase 'can help'.
**Reply:** Rephrased in "can facilitate the incorporation of litter-derived carbon"

P 1, L 9: Add that wild boar bioturbation was simulated.
**Reply:** We added that bioturbation was simulated

P 1, L 17: Please rephrase 'can help'.
**Reply:** Rephrased in: "Wild boar may speed up this process with their grubbing activity."

Introduction:
P 1, L 30: I suggest either to replace 'the main process' by 'a major process' or to add
an appropriate reference to that statement.

**Reply:** We changed this accordingly

P 3, L 2: Please add the references of the studies which have investigated wild boar
effects on soil carbon stocks.
**Reply:** We will add the references to Wirthner, 2011 and  Mohr and Topp, 2001 here.

P 3, L 11: Add that the effects of 'simulated' wild boar grubbing were investigated.

**Reply:** "Simulated" will be added.

Materials and Methods
P 3, L 16, 17: I think there should be an 'a' before mean annual temperature and moisture.
**Reply:** Will be added.

P 4, L 14-19: It took me a while to understand the idea behind mass equivalent sampling, and why you applied it in this study. However, I'm still not sure if I properly understand it and I'm therefore a bit concerned if this procedure might affect the results. By sampling the same amount of soil per horizon and pit, I have the feeling you could underestimate potential C losses from bioturbation. For example, in the theoretical case, 50% of the LF horizon organic matter stocks would have been mineralized due to bioturbation, this sampling procedure would artificially 'refill' the missing amount of organic matter with organic matter from the next horizon (i.e. O horizon). Now, the O horizon is (artificially) smaller, but will be 'refilled' with soil from the next layer, and so on. At some point, material from a deeper layer which has not been sampled at the reference plot would be sampled to 'refill' the missing amount of organic matter. Thus, the actual amount of lost C would be underestimated. I might be completely wrong, but then I suggest to elaborate more in detail on the sampling procedure.

**Reply:** We agree that the sampling procedure is not easy to understand. Therefore, we included the figure for illustration and we will revise this paragraph where necessary to make more clear. However, the general concept was developed by Ellert and Bettany 1995 and before also by Jenkinson (see also Wendt and Hauser 2013, EJSS). In most cases it is recommend applying a mass correction to the obtained soil data set. However, sampling directly in a mass corrected way is the preferable method to correct for differences in soil mass. The "refilling" of missing soil mass the reviewer are referring to, is done with subsoil material that is C poor. Thus, the sampling procedure can hardly bias results in a way outlined by the reviewer: If O horizon material (around 500 g/kg Corg) would be lost due to bioturbation, it would be replaced by subsoil material with 2.5 g/kg Corg in an equivalent soil mass sampling. Thus, a 10.00 % loss of O material would result in a 9.95% C-loss with the refilling from subsoil material. This is practically the same C-loss and the error is far below the precision that can be achieved with any soil sampling. We agree, that sampling the soil treatments is never completely without bias but, as explained above, the sampling procedure will not bias the results in a way that our interpretation is not valid anymore.

P 5, L 1-2: Was there only one composite sample per site, treatment and soil horizon? Please clarify.
**Reply:** We will add that it was one composite sample per site and treatment.

P 5, L 14: Which statistical test did you use? What was you level of significance? How, did you account for nesting within sites? Please clarify.
**Reply:** We will add the missing information (mixed linear model (lme function, nlme package) accounting for site and area as nested random factors ($\alpha$=0.05)) with the following sentences: " The differences in SOC stocks were analysed using mixed linear models (package nlme, function lme) accounting for site and area as nested random factors ($\alpha$=0.05). Tukey's honest significant difference post-hoc test was applied."

Results
P 5, L 17: Again, please indicate that bioturbation was simulated (here and elsewhere in the text).
**Reply:** We will add "simulated"

P 5, L 18: Fig. 2a not 2A.
**Reply:** Will be corrected

P 5, L 23: In my opinion it is not necessary to show the results of the individual plots. Thus I would suggest to move Fig. 3 to the supplements.
Reply: We agree and shift this figure to a supplement.

P 6, L 4: Instead of showing the individual plots (Fig. 3) I suggest to add a figure showing the bioturbation effect separate for the forest types.
**Reply:** As it is visible in Fig. 3, the variability between the sites is large and there are no differences between the forest types. The Fig. 3 is designed with four lines that comprise the different forest types in the two areas.

P 6, L 8-10: This should be part of the discussion.
**Reply:** We revised this to "This was related to greater thickness of the forest floor and a larger proportion of mineral soil SOC in deciduous and coniferous forest reference plots than in mixed forest plots." and would argue that this is still a result and no interpretation.

P 6, L 14: It is stated earlier that fractions were determined on composite samples only. How did you do statistics on that? Please clarify.
**Reply:** Statistics was conducted not at site level but taking all sites and both areas into account (see above).

P6, L 15: Clarify that 'treatment effects' were similar among forest types. In the present form I first thought that e.g. POM fractions were similar among forest types.
**Reply:**

P 6, L 19: is 'total stocks of MOM' correct? Or should it be MOM fraction? This reads a bit confusing. I would also suggest to refer to Fig. 4a here.
**Reply:** Thank you for noticing. We will rephrase into "MOM fraction) and refer to Fig 4a and 4b.

P 6, L 23-24: Please add the forest floor POM/SOM proportion and stocks to Figure 4.
**Reply:** A fractionation of SOM into POM and MOM for the forest floor is technically not possible due to the very low fraction of minerals in the forest floor. We will add this in the figure legend. It can be approximated that almost all SOM in the forest floor is POM.

Although this results are included in Fig. 5, I think it would be more clear if you add it to Fig. 4.
**Reply:** See above.

P 7, L 4: Was there no mineral surface C in the forest floor of the reference plots? Did
you measure it? Please clarify.
**Reply:** The forest floor had a carbon content of almost 50% with is equal to almost pure organic matter.
With the applied fractionation procedure it was not possible to fractionate this material with almost no
minerals.

Discussion:
P 8, L 30: I guess it should be Fig. 2b not 1b.
**Reply:** Thank you for noticing. It will be corrected.

P 9, L 10: Please cross-check figure reference.
**Reply:** Thank you for noticing. It will be corrected.

Conclusion
P 10, L 16: Please, add that wild boar activity was simulated.
**Reply:** Will be added.

To put your results into a bigger context, I suggest to add some information/thoughts
about potential long-term consequences.
**Reply:** We added the following sentence: "On long-term this may even lead to enhanced SOC stocks due
to an increased fraction of stabilised SOC. Soil disturbance with mixing and bioturbation were previously
assumed to enhance SOC mineralisation and cause SOC losses. This could not be confirmed in our study
and calls for a new perception. Soil mixing with bioturbation or anthropogenic with machinery lead to a
more even distribution of SOC in the soil profile and may result in enhanced SOC stocks on long-term."

Figures:
Figure 2: In the case horizons showed significant differences between treatments
please indicate that by adding significance stars to the figure.
**Reply:** We will include the information on the significance between the horizons in the figure captions
since it is difficult to visualize between the bars: "Significant differences were found in 0-5 cm depth and
in the combined forest floor (L+O layer)."

Figure 3: This figure should be moved to the supplements. Instead replace it with a
barchart for each forest type. Add significance stars to the figure.
**Reply:** This figure illustrates the variability between the plots, but we agree that it can be shifted to the
supplement. A figure with the forest types can be added, but it may not be useful since we found no
significant differences between the forest types.

Figure 7: In the case horizons showed significant differences between treatments,
please indicate that by adding significance stars to the figure. What happened to L+O
of the reference plots?

**Reply:** This figure refers to the MOM-fraction. There was no mineral fraction in the L+O-horizon in the
reference plots and therefore this fraction could not be analysed for the reference treatment.

---

## Author Comment (AC2) · 17 Jun 2019

**Reply to reviewer 2**

General comments:
Axel Don et al. have submitted an original, well written and very interesting draft to BG.
They investigated the effect of wild boar bioturbation on soil organic carbon (SOC)
dynamics in a 6 year study in two forests of Germany (focusing on coniferous plots in
one forest and coniferous + deciduous plots in the second forest), both on acidic and
sandy soils.
The experimental design is nice and sound, the authors have manually simulated wild
boar bioturbation (each year, which is a high frequency). They discuss in a clear way the advantages and
limitations of the chosen design. Yet, I think that in many sentences
of the manuscript and also in the draft title, the fact that wild boar were not part of the
game should be more clearly stated (see below my specific comment 1 on this topic).
**Reply:** We very much appreciate all comments and suggestions from the reviewer and took all of them
into account. We agree and added "simulated" wherever wild boar bioturbation is mentioned.

SOC dynamics is studied in paired-plots (control vs. bioturbation) by focusing on SOC
stocks (using equivalent soil mass for litter + 0-5/5-10/10/15 cm soil layers) and SOC
physical fractionation to separate particulate SOC from mineral associated SOC (0-
5/5-10 cm soil layers + C content of the mineral fraction of the O layer of bioturbed
plots). The results show that SOC stocks were not affected by bioturbation but that the
fate of litter SOC was affected by bioturbation: 1/ a part remains in the litter (as litter),
2/ a part is incorporated as particulate/light organic matter in the mineral topsoil layer,
3/ a part remains in the litter layer, but is associated to minerals.
The authors finally state that the part of the litter SOC that has been associated to
minerals (in the litter layer) has been "stabilised" by bioturbation. I suggest that this
statement on carbon "stabilisation" should be avoided. We indeed lack evidences regarding
the residence time of mineral-associated SOC above the topsoil (i.e. in the
litter layer; see below my specific comment 2 on this topic).
**Reply:** See reply below.

Specific comments:
1/ Wild boars were not involved in this study :)
I suggest to state more clearly in the title and in the text that bioturbation by wild boar
was simulated. - "Simulated wild boar bioturbation..." for the title. - in the text this could
be done for instance p5 line 17 "[simulated] wild boar bioturbation" and in many other
sentences of the draft.
**Reply:** We agree and added "simulated" wherever wild boar bioturbation is mentioned.

2/ Mineral-associated C in the litter layer (above-ground) cannot be called "stabilised" C
First, I would like to remind that in (mineral) soils, a large part of mineral-associated
SOC is not stabilised. This has been clearly shown in e.g. long term bare fallows trials where the fine soil
fraction looses SOC at a relatively high rate. So transfering litter
SOC to the mineral-associated SOC fraction does not mechanistically imply that all of it
has been stabilised. A part of it may be stabilised if this transfer would have taken place
in the mineral soil layer (i.e. below the soil surface). Indeed the mean residence time
of mineral-associated SOC is generally higher that the one of the particulate organic
matter SOC in mineral soil layers. But here the bioturbation transfer of SOC from litter to

the mineral-associated C fraction occurred in the litter layer (i.e. above the soil surface), where there is no evidence that this above-ground mineral-associated C would have a slower turnover than litter C from the F/H O layers. This should be acknowledged in the manuscript.

**Reply:** We will add in the material and method section to define the term "stabilised" by adding the following sentence: We refer to SOC associated with minerals (MOM) as stabilised SOC since its turnover is slower compared to non-mineral associated POM. "Stabilised" does not mean "inert" but only refers to the fact that organic compounds attached to mineral surfaces are more difficult for microorganisms to use as substrate. There is a large number of studies showing that under different environmental conditions (also in aquatic systems) the mineral association of organic compounds reduces its turnover (e.g. Eusterhues et al. 2003 Organic Geochemistry, Six et al. 2002, Plant and Soil, Kleber et al. 2007 Biogeochemistry, von Lützow et al. 2006, European Journal Soil Science). Bioturbation leads to a complete mixture of organic layer and mineral soil and it is not possible to distinguish both compartments anymore. Moreover, in forests with intensive bioturbation (by earthworm) only temporarily forest floor can be found and litter is incorporated into the mineral soil. This is a similar situation like in the investigated plots with simulated bioturbation. Thus, from an ecological and biogeochemistry point of view there is no reason to assume that attachment of organic compounds will not lead to stabilisation in the sense to decreased turnover.

- The title of the manuscript should be changed, avoiding the confusing term (and not properly measured for litter layers) "stability". The expression "increases mineral C loading" should be preferred and would better represent the findings of the study.
**Reply:** We do not agree that the term "increased mineral C loading" will be more clear and easy to understand compared to "increases the stability of forest soil carbon". Here we only refer to a gradient change in stability/turnover. Thus, this is not misleading but reflects the fact that more SOC is incorporated into the mineral soil making it less prone to disturbances such as forest fires and also more SOC is mineral associated (see above).

- The title of section 2.2 should be changed. Stability of SOC was not assessed, but "SOC distribution in physical soil fractions".
**Reply:** We will change the title accordingly into "Distribution of organic carbon in physical soil fractions"

- The title of section 2.3 should be changed to "Associating C on minerals with bioturbation".
Reply: We propose to delete this heading and include the section into the previous one with an introductory sentence.

- The title of section 3.2 should be changed to "Contact [...] facilitates the association of litter C with minerals in the litter layer"
**Reply:** We will revise the heading and deleted the word "soil" in order to emphasis the transitional character of the new compartment in which litter layer and mineral soil is mixed together. However, as explained above, we do not see any reason to assume that mineral association of organic carbon does not lead to slower turnover and thus higher C stability. We propose as new heading "Contact between litter layer carbon and minerals facilitates carbon stabilisation".

- The abstract/conclusion should be re-written: bioturbation has a positive effect on "C association to minerals" or on "mineral C loading in the litter layer", not on C stability, we do not know of this C is "more stabilised", it is more linked to minerals, and research on the turnover of mineral-associated C in the litter layer is therefore needed.
**Reply:** See reply above.

3/ The effect of wild boar on plant biodiversity and forest ecosystem C cycle is questionable
In the introduction section, the authors insist on the "mainly positive effects in forests" (p2 line 15) of wild boar bioturbation. However, other studies are questioning this statement, presenting the effect of ungulate populations as :
- "jeopardiz[ing] forest regeneration process"
- "detrimental to the peculiarity of forest plant communities"
- leading to "lanscape-level biotic homogeneization" (see e.g. Boulanger et al., 2017 in Global Change Biology)
If forest regeneration process is actually jeopardized by wild boar invasions, then the fate of the global ecosystem C stock and cycle is not clear... This should be acknowledged in the manuscript.
Reply: We acknowledge that the effects of wild boar on biogeochemical processes and forest ecology are not fully understood and may also be negative.
**Reply:** We agree that studies on wild boar effects are not uniform in their results. Therefore, we will include a sentence acknowledging that wild boar may also have negative effects: "However, other studies also found negative wild boar effects on forest regeneration or understory biodiversity (Siemann et al., 2009; Barrios-Garcia and Ballari, 2012)". Since Boulanger et al., 2017 did not find any effect of wild boar on species richness for any vegetation layer, we refrain from citing this paper as example for negative wild boar effects.

4/ No positive grubbing effect on total SOC stocks were found
Please correct this mistake at p8 lines 9-10
**Reply:** We will correct this and change it to: "supporting our finding of no SOC loss with bioturbation".

Technical corrections:
p1 l26: "an[d]" ?
**Reply:** Thank you for noticing. It will be corrected.

p5 l17: "due [to] six"
**Reply:** Thank you for noticing. It will be corrected.

p7 l2: "significant[ly]
**Reply:** Will be changed accordingly.

p8 l20: please replace "mainly" with "only"
**Reply:** Will be changed accordingly.

p9 l16: please reverse "forest floor" and "mineral soil" : "mineral soil" : mineral soil mixed into the forest floor almost doubled the C load (not the opposite)
**Reply:** Will be changed accordingly.